# Bridging the Gap Between Zeroth-Order and First-Order Fine-Tuning via Dynamic Adaptive Subspace Pre-tuning

## Abstract

Fine-tuning large language models (LLMs) faces a trade-off between the accuracy of first-order (FO) methods and the memory efficiency of zeroth-order (ZO) optimizers. While ZO methods avoid the activation memory bottleneck of backpropagation, they typically converge slowly and show a noticeable performance gap compared to FO approaches. To address this, we propose **Dynamic Adaptive Subspace Pre-tuning (DASP)**, a framework that combines the efficiency of ZO methods with the accuracy of FO methods. DASP introduces a lightweight pre-computation stage that constructs low-rank, layer-wise subspaces aligned with the loss landscape. Fine-tuning is then restricted to small transformation matrices within these fixed subspaces, greatly reducing optimizer state memory. To further eliminate activation memory overhead, DASP employs a streaming backpropagation algorithm that decouples peak memory from sequence length. Experiments on LLaMA3 and OPT-13B show that DASP consistently outperforms ZO baselines by large margins (e.g., +6.5% on RTE with LLaMA3), while matching the accuracy of FO methods at even lower memory cost. These results highlight DASP as a practical and scalable solution for memory-efficient LLM adaptation.

## 1 Introduction

Adapting large language models (LLMs) to downstream tasks is constrained by a trade-off among performance, memory footprint, and computational cost (Chen et al., 2024b; Zhang et al., 2024). First-order (FO) fine-tuning delivers strong accuracy but requires storing optimizer states, gradients, and, most critically, activations for backpropagation (Kingma & Ba, 2017; Chen et al., 2016), leading to prohibitive memory usage. Parameter-efficient fine-tuning (PEFT) methods such as LoRA (Hu et al., 2022) alleviate some of these costs by introducing low-rank updates on weights, but they do not address the activation memory bottleneck, which scales with sequence length and model size.

Zeroth-order (ZO) optimization provides an alternative by eliminating backpropagation and thus removing activation storage (Malladi et al., 2024a). However, ZO methods converge slowly, require many forward passes, and show a consistent accuracy gap compared to FO methods. A key reason is that ZO optimizers rely on high-rank random probes, which fail to exploit the empirically low-rank structure of gradients in LLM adaptation (Zhao et al., 2024b; Hao et al., 2024). As a result, current approaches force a choice between high-performance but memory-intensive FO fine-tuning and memory-efficient but underperforming ZO optimization, with no method reconciling these trade-offs.

We show that this trade-off is not inherent. To resolve it, we propose **Dynamic Adaptive Subspace Pre-tuning (DASP)**, a fine-tuning framework that achieves FO-level performance at lower memory cost than ZO methods. The key insight is to decouple the discovery of an adaptation subspace from the optimization within it.

DASP implements this idea through a two-stage design. Figure 1 illustrates the overall DASP framework. The first stage is an efficient, ZO-inspired **offline pre-computation**. Unlike LoRA, which starts from random projections, DASP uses an iterative probing algorithm to identify a shared low-rank "sensitive subspace" that captures the main directions of change for a given pre-trained model. An important property of this stage is its **transferability**: the computed subspace bases ($P$ and

Figure 1: **The DASP framework: a two-stage paradigm for memory-efficient LLM fine-tuning.** Conventional zeroth-order (ZO) methods (left) directly optimize in the parameter space and suffer from high cost and low efficiency. In contrast, DASP (right) decouples *subspace discovery* from *optimization*: (1) an **offline pre-computation** stage discovers a low-rank "sensitive subspace" ($P$, $Q$); (2) an **online fine-tuning** stage updates only a small core matrix ($\delta$) within this subspace, using Flow Backpropagation (FBP) to enable exact long-context training with constant activation memory.

$Q$) can be reused across different downstream tasks, amortizing the one-time cost. The second stage is an **online fine-tuning** step that is both lightweight and memory-efficient. With the subspace fixed, fine-tuning reduces to optimizing a small rank-dimensional transformation matrix ($\delta \in \mathbb{R}^{r \times r}$), which keeps optimizer and gradient memory overhead negligible and more efficient than LoRA.

To further address activation memory, we develop **F**low **B**ack**p**ropagation (FBP), an exact streaming backpropagation algorithm tailored for DASP. FBP compresses memory in two dimensions: (1) it processes the backward pass in sequential chunks, decoupling peak memory from sequence length ($T$); and (2) by leveraging pre-computed projections, it reduces hidden-state memory along the model dimension ($d$). This projection-aware mechanism allows fine-tuning with near-constant activation cost, even for long contexts. Figure 2 summarizes the memory footprint (a), training loss (b), and accuracy (c) of DASP versus common PEFT and ZO baselines. DASP establishes a new state-of-the-art in efficient fine-tuning, achieving a memory footprint substantially lower than LoRA and comparable to MeZO, while converging faster to a superior loss value and delivering higher downstream task performance at a fraction of the computational cost.

Our contributions are as follows:

- We propose DASP, a framework that decouples subspace discovery from optimization via a transferable pre-computation stage, significantly reducing optimizer and gradient memory.
- We introduce FBP, a projection-aware backpropagation algorithm that compresses activation memory along both sequence and hidden dimensions, enabling exact long-context fine-tuning with near-constant memory usage.
- We show that DASP matches FO accuracy while surpassing state-of-the-art PEFT and ZO methods in memory efficiency, providing a practical solution for scalable LLM adaptation.

## 2 RELATED WORK

**Parameter-Efficient Fine-Tuning (PEFT).** To mitigate the prohibitive costs of full-parameter fine-tuning, a diverse set of PEFT methods has been proposed. These methods adapt pre-trained models by training only a small fraction of parameters. Prominent among them is Low-Rank Adaptation (LoRA) (Hu et al., 2022), which injects trainable, low-rank decomposition matrices into the layers of a frozen base model, thereby drastically reducing the memory required for gradients and optimizer states. Other paradigms include adapter-based methods that insert small bottleneck mod-

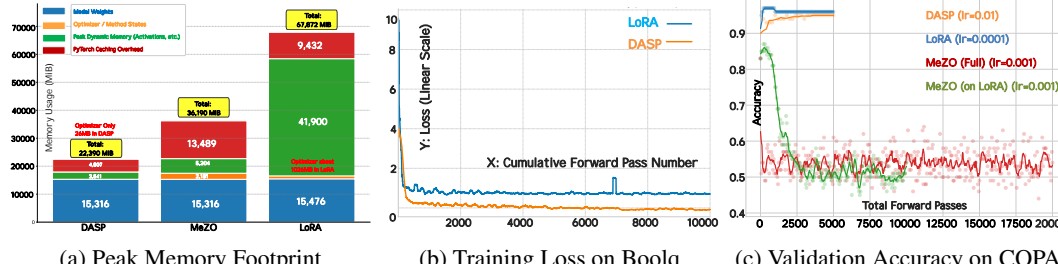

(a) Peak Memory Footprint  (b) Training Loss on Boolq  (c) Validation Accuracy on COPA

Figure 2: **Comparison of DASP against LoRA and ZO baselines.** **(a)** DASP drastically reduces peak memory, achieving a footprint comparable to MeZO (ZO) by eliminating the activation bottleneck that inflates LoRA's memory usage. **(b)** DASP efficiency translates to faster, more stable convergence to a lower training loss than LoRA. **(c)** DASP achieves higher final validation accuracy in significantly fewer forward passes, outperforming both LoRA and ZO methods.

ules (Houlsby et al., 2019), and prompt-based methods such as prompt-tuning (Lester et al., 2021) and prefix-tuning (Li & Liang, 2021), which optimize continuous prompt embeddings prepended to the input. While effective at lowering optimizer and gradient memory, these methods still require a full backward pass, leaving the activation memory bottleneck unaddressed.

**Memory-Efficient Backpropagation.** To address the activation footprint, another line of research focuses on memory-efficient backpropagation. The most established technique is gradient checkpointing (Chen et al., 2016), which reduces memory by re-computing activations during the backward pass instead of storing them. However, it still incurs a large peak memory cost when the activations for each checkpointed layer are materialized. More recent work has sought to overcome this limitation. For instance, StreamBP (Luo et al., 2025) and MsT (Zhao et al., 2024a) partition the sequence and stream the backward pass, thereby avoiding the need to store the full activation tensor at once. These approaches highlight the growing emphasis on activation memory reduction, but their efficiency still scales with sequence length.

**Zeroth-Order (ZO) Optimization for LLMs.** A more radical alternative circumvents backpropagation entirely via zeroth-order optimization. By estimating gradients only from function evaluations (forward passes), methods such as MeZO (Malladi et al., 2024a) eliminate the need to store activations, making them attractive for memory-constrained environments. However, this comes at the cost of convergence and accuracy: ZO methods typically require many forward passes and lag behind FO-based methods in final performance. Recent attempts to narrow this gap include leveraging second-order information (Zhao et al., 2025), exploiting gradient sparsity (Liu et al., 2024), or enforcing low-rank structures in the gradient estimate (Chen et al., 2024a). Despite these improvements, the trade-off between ZO's extreme memory efficiency and FO's high performance remains unresolved.

In summary, PEFT reduces optimizer memory but not activations, memory-efficient backpropagation alleviates but does not eliminate activation bottlenecks, and ZO methods avoid activations altogether but suffer from significant performance deficits. This trade-off motivates our DASP framework, which simultaneously achieves FO-level accuracy and memory efficiency beyond ZO.

## 3 METHOD

### 3.1 MOTIVATION AND INSIGHT

A key limitation of existing fine-tuning paradigms is that they conflate two distinct processes: *finding* an appropriate low-dimensional subspace and optimizing within it. LoRA assumes a low-rank structure but starts from randomly initialized subspaces, requiring substantial training to align them. ZO methods disregard this structure altogether, wasting probes on irrelevant directions. Our central insight is that these processes can and should be **decoupled**.

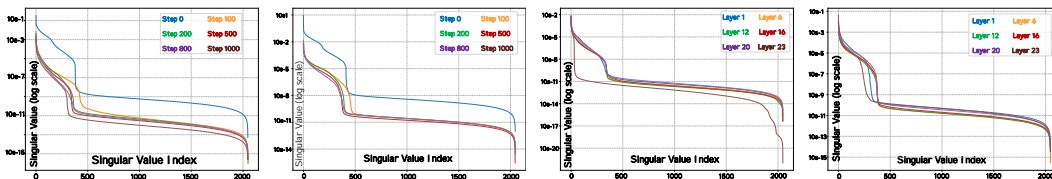

(a) Temporal: Q Proj (L6)  (b) Temporal: V Proj (L6)  (c) Spatial: Q Proj (S500)  (d) Spatial: V Proj (S500)

Figure 3: **Analysis of the low-rank structure of gradients during fine-tuning OPT-1.3B on SST-2.** The singular value spectra of gradient matrices exhibit a consistent sharp decay, indicating a persistent low-rank structure. This property holds **(a, b) temporally**, across different training steps for a fixed layer (Layer 6), and **(c, d) spatially**, across different layers at a fixed step (Step 500).

Figure 3 shows the singular value spectrum of gradient matrices from OPT-1.3B during fine-tuning. The spectra consistently exhibit sharp decay, confirming that gradients are intrinsically **low-rank**. This property is stable both **across training steps** (Figs. 3a, 3b) and **across layers** (Figs. 3c, 3d). In other words, adaptation primarily occurs within a compact subspace, while most parameter dimensions remain unused.

This observation explains the inefficiency of prior methods: LoRA learns the subspace on the fly, while ZO wastes effort exploring the null space. It also suggests a more efficient alternative: if a stable "sensitive subspace" exists, it can be identified before fine-tuning. Moreover, its consistency across time and layers implies that it is a property of the model itself, rather than any specific dataset. We therefore hypothesize that the sensitive subspace is **transferable**. By investing in a one-time, offline pre-computation to discover this subspace, we can amortize the cost and enable subsequent fine-tuning to be extremely efficient. DASP instantiates this paradigm by first discovering the universal subspace, and then optimizing solely within it.

### 3.2 PRELIMINARIES: SUBSPACE-CONSTRAINED ADAPTATION

PEFT methods such as LoRA constrain the weight update $\Delta W$ of a linear layer $W \in \mathbb{R}^{m \times n}$ to a low-rank form:

$$W' = W + \Delta W = W + BA, \tag{1}$$

where $B \in \mathbb{R}^{m \times r}$ and $A \in \mathbb{R}^{r \times n}$ are trainable low-rank matrices, with rank $r \ll \min(m, n)$.

DASP adopts a similar formulation but makes a critical shift. The update is parameterized by two fixed, orthonormal bases $P \in \mathbb{R}^{m \times r}$ and $Q \in \mathbb{R}^{n \times r}$, and a tiny trainable core matrix $\delta \in \mathbb{R}^{r \times r}$:

$$W' = W + P\delta Q^T. \tag{2}$$

Here, $P$ and $Q$ define the "*sensitive subspace*", discovered once in a pre-computation stage and then frozen. Fine-tuning is reduced to updating only $\delta$, making optimizer and gradient storage negligible compared to LoRA.

### 3.3 STAGE 1: OFFLINE PRE-COMPUTATION OF SENSITIVE SUBSPACES

The quality of DASP relies on discovering subspaces $(P, Q)$ that are maximally sensitive to perturbations with respect to task loss. This is achieved through a ZO-inspired iterative procedure that requires only forward passes.

**Objective.** We aim to learn orthonormal bases $(P, Q)$ that maximize loss deviation under random, rank-$r$ perturbations. Formally, for a loss $\mathcal{L}$ and calibration dataset $\mathcal{D}_{cal}$:

$$\max_{P^T P = I, Q^T Q = I} \mathbb{E}_{x \sim \mathcal{D}_{cal}, \omega \sim \mathcal{N}(0, I_{r \times r})} \left[ |\mathcal{L}(W + \epsilon P \omega Q^T; x) - \mathcal{L}(W; x)| \right], \tag{3}$$

where $\epsilon$ is a small perturbation scalar and $\omega$ is a random core matrix.

**Iterative Algorithm.** Solving Eq. 3 directly is intractable. We thus adopt an iterative refinement scheme. Starting from random bases $(P_0, Q_0)$, each iteration $k$ generates several random probes

---

**Algorithm 1** The DASP Framework

---

1: **Offline Stage: Pre-compute Subspace Bases** $(P, Q)$
2: **Input:** Pre-trained model $M$ with weights $\{W_l\}$, calibration dataset $\mathcal{D}_{cal}$, rank $r$.
3: **for all** linear layer $W_l \in M$ **do**
4:     Initialize random orthonormal bases $P_{l,0}, Q_{l,0}$.
5:     **for** $k = 0$ to $K - 1$ **do**
6:         Sample data batch $x \sim \mathcal{D}_{cal}$.
7:         Find best probe $\omega^* = \arg\max_{\omega_j} |\mathcal{L}(W_l + \epsilon P_{l,k}\omega_j Q_{l,k}^T; x) - \mathcal{L}(W_l; x)|$.
8:         $U, \_, V^T \leftarrow \text{SVD}(\omega^*)$.
9:         $P_{l,k+1} \leftarrow \text{orth}(P_{l,k}U), Q_{l,k+1} \leftarrow \text{orth}(Q_{l,k}V)$.
10:     **end for**
11: **end for**
12: **Return** bases $\{P_l, Q_l\}$.
13: **Online Stage: Fine-tune Core Matrices** $(\delta)$
14: **Input:** Model $M$, pre-computed bases $\{P_l, Q_l\}$, downstream dataset $\mathcal{D}_{task}$.
15: Initialize trainable parameters $\{\delta_l \in \mathbb{R}^{r \times r}\}$ with zeros. Freeze $M, \{P_l, Q_l\}$.
16: **for all** training step **do**
17:     Sample batch $(x, y) \sim \mathcal{D}_{task}$.
18:     Compute loss $\mathcal{L}(M, \{\delta_l\}; x, y)$ via forward pass incorporating Eq. 2.
19:     Compute gradients $\{\nabla_{\delta_l}\mathcal{L}\}$ using Flow Backpropagation (FBP) via Eq. 15.
20:     Update $\{\delta_l\}$ using an FO optimizer (e.g., AdamW).
21: **end for**

---

$\{\omega_j\}_{j=1}^{N_{probes}}$ and selects $\omega^*$ that induces the largest loss change. Decomposing $\omega^*$ via SVD ($\omega^* = U\Sigma V^T$) yields rotations $U$ and $V$, which are applied to update and re-orthogonalize the bases:

$$P_{k+1} = \text{orth}(P_k U) \tag{4}$$

$$Q_{k+1} = \text{orth}(Q_k V) \tag{5}$$

where $\text{orth}(\cdot)$ denotes an orthogonalization step (e.g., QR decomposition). As summarized in Algorithm 1, this iterative refinement gradually steers the bases toward the principal axes of maximal loss curvature. A key feature is their **transferability**: once computed, these task-agnostic subspaces can be serialized as a reusable asset, eliminating repeated computation across downstream tasks.

### 3.4 STAGE 2: ONLINE FINE-TUNING WITH FLOW BACKPROPAGATION (FBP)

With the sensitive subspace bases $P$ and $Q$ fixed, the online objective reduces to learning the small core matrix $\delta$:

$$\min_{\delta} \mathcal{L}(W + P\delta Q^T). \tag{6}$$

This drastically lowers the number of trainable parameters. However, naïve backpropagation still suffers from the activation memory bottleneck for long sequences. To overcome this, we introduce **Flow Backpropagation (FBP)**.

**Gradient Derivation for Core Matrix** $\delta$**.** The DASP adapter modifies the output of a linear layer $Y = XW^T$ by adding a low-rank term. The adapted output $Y'$ is given by:

$$Y' = Y + X(P\delta Q^T)^T = Y + XQ\delta^T P^T, \tag{7}$$

where $X \in \mathbb{R}^{T \times d}$ is the input activation, and $P \in \mathbb{R}^{m \times r}, Q \in \mathbb{R}^{n \times r}, \delta \in \mathbb{R}^{r \times r}$ are the pre-computed bases and the trainable core matrix, respectively. Our goal is to derive the gradient of the loss $\mathcal{L}$ with respect to $\delta$. We present a formal derivation using the properties of matrix calculus and the trace operator.

*Proof of Gradient Formula.* Let $G = \nabla_Y \mathcal{L} \in \mathbb{R}^{T \times m}$ be the gradient of the loss propagated back to the output of the layer. The differential of the loss $d\mathcal{L}$ can be expressed as the trace of the inner product between the gradient and the differential of the weights:

$$d\mathcal{L} = \text{tr}((\nabla_{W'}\mathcal{L})^T dW') = \text{tr}(G^T X(d(P\delta Q^T))^T). \tag{8}$$

Since $P$ and $Q$ are frozen, the differential is only with respect to $\delta$. Using the property $(ABC)^T = C^T B^T A^T$, we have:

$$d(P\delta Q^T) = P(d\delta)Q^T \implies (d(P\delta Q^T))^T = (Q^T)^T (d\delta)^T P^T = Q(d\delta)^T P^T. \tag{9}$$

Substituting this back into the expression for $d\mathcal{L}$:

$$d\mathcal{L} = \mathrm{tr}(G^T XQ(d\delta)^T P^T). \tag{10}$$

Leveraging the cyclic property of the trace operator, $\mathrm{tr}(ABCD) = \mathrm{tr}(DABC)$, we can rearrange the terms to isolate $d\delta$:

$$\begin{aligned} d\mathcal{L} &= \mathrm{tr}(P^T G^T XQ(d\delta)^T) \\ &= \mathrm{tr}(((XQ)^T(GP))^T (d\delta)^T), \end{aligned} \tag{11}$$

where we use the identity $(AB)^T = B^T A^T$. The final expression is in the form $d\mathcal{L} = \mathrm{tr}((\nabla_\delta \mathcal{L})^T d\delta)$, from which we can directly identify the gradient:

$$\nabla_\delta \mathcal{L} = (XQ)^T(GP) = (Q^T X^T)(GP). \tag{12}$$

This formally derives the gradient for the core matrix $\delta$.

**Memory Analysis and the Remaining Bottleneck.** Eq. 12 suggests a natural path to memory savings: the gradient can be computed via the projected matrices $X_p = XQ \in \mathbb{R}^{T \times r}$ and $G_p = GP \in \mathbb{R}^{T \times r}$. This reduces the hidden-dimension memory footprint from $O(T \cdot d)$ to $O(T \cdot r)$, a substantial improvement since $r \ll d$.

However, this projection only addresses one side of the problem. The memory cost still grows linearly with the sequence length $T$, making $O(T \cdot r)$ prohibitive for long-context settings where $T$ can reach tens of thousands. Thus, despite the hidden-dimension compression, the sequence-length dependence reintroduces an activation memory wall. Eliminating this $T$—scaling requires a more advanced mechanism—precisely the motivation for our FBP mechanism.

**FBP Mechanism.** A naive evaluation of Eq. 12 requires materializing the intermediate matrices $XQ \in \mathbb{R}^{T \times r}$ and $(\nabla_Y \mathcal{L})P \in \mathbb{R}^{T \times r}$, both of which scale linearly with the sequence length $T$, reintroducing the activation memory wall. FBP circumvents this by leveraging the fundamental properties of block matrix multiplication, which guarantees that a partitioned, streaming computation is mathematically identical to the full matrix computation.

Inspired by StreamBP (Luo et al., 2025), our FBP similarly leverages the principle of linearly decomposing the backpropagation chain rule along the sequence dimension. The core innovation of FBP, is to apply this principle within the unique low-rank structure of our DASP framework. All memory-efficient computations thus occur entirely within the $r$-dimensional sensitive subspace spanned by the pre-computed bases $P$ and $Q$. This design ensures that the computational complexity and peak memory overhead of FBP are only proportional to the chunk size $C$ and the small rank $r$, remaining independent of the model's large hidden dimension $d$.

*Proof of Equivalence.* Let the input activations $X \in \mathbb{R}^{T \times d}$ and the incoming output gradients $G = \nabla_Y \mathcal{L} \in \mathbb{R}^{T \times m}$ be partitioned into $k$ disjoint chunks along the sequence dimension (rows), where each chunk has size $T_i$ such that $\sum_{i=1}^k T_i = T$. We can express $X$ and $G$ as block matrices:

$$X = \begin{bmatrix} X_1 \\ X_2 \\ \vdots \\ X_k \end{bmatrix}, \quad G = \begin{bmatrix} G_1 \\ G_2 \\ \vdots \\ G_k \end{bmatrix}, \tag{13}$$

where $X_i \in \mathbb{R}^{T_i \times d}$ and $G_i \in \mathbb{R}^{T_i \times m}$. The full projected matrices can thus be written as:

$$XQ = \begin{bmatrix} X_1 Q \\ X_2 Q \\ \vdots \\ X_k Q \end{bmatrix}, \quad GP = \begin{bmatrix} G_1 P \\ G_2 P \\ \vdots \\ G_k P \end{bmatrix}. \tag{14}$$

Now, we evaluate the full gradient formula from Eq. 12 using these block matrix forms:

$$\nabla_\delta \mathcal{L} = (GP)^T(XQ) = \begin{bmatrix} (G_1P)^T & (G_2P)^T & \cdots & (G_kP)^T \end{bmatrix} \begin{bmatrix} X_1Q \\ X_2Q \\ \vdots \\ X_kQ \end{bmatrix}$$

$$= (G_1P)^T(X_1Q) + (G_2P)^T(X_2Q) + \cdots + (G_kP)^T(X_kQ)$$

$$= \sum_{i=1}^{k}(G_iP)^T(X_iQ). \tag{15}$$

The final expression in Eq. 15 shows that the gradient can be written as a summation of chunk-wise partial terms $(G_iP)^T(X_iQ)$. Formally, this establishes that the chunked summation is *mathematically identical* to the gradient obtained with the full unpartitioned matrices, i.e., FBP introduces no approximation or precision loss.

FBP realizes this equivalence in practice by iterating through sequential chunks during backpropagation: for each chunk $i$, it computes $(G_iP)^T(X_iQ)$, accumulates the result into $\nabla_\delta \mathcal{L}$, and discards the intermediate $X_i$ and $G_i$. This streaming scheme decouples the peak activation memory from the sequence length $T$. Moreover, since the projections $X_iQ$ and $G_iP$ reside in the low-dimensional subspace of size $r \ll d$, FBP also reduces the hidden-dimension memory footprint.

## 4 EXPERIMENTS

### 4.1 EXPERIMENTAL SETUP

**Models.** We evaluate DASP on a diverse set of modern, publicly available LLMs to highlight its general applicability. The suite includes decoder-only models (**Llama3-8B**, **Phi-2 (2.7B)**, and the **OPT** family up to 13B) as well as the encoder-based **RoBERTa-large** (350M).

**Datasets.** Experiments are conducted on a broad range of tasks from **GLUE** and **SuperGLUE**, covering classification (**SST-2**, **SST-5**, **RTE**, **CB**, **BoolQ**, **WSC**, **WIC**, **TRECC**, **MultiRC**), natural language inference (**SNLI**, **MNLI**), and multiple-choice reasoning (**COPA**). This ensures a comprehensive assessment across language understanding, reasoning, and inference.

**Baselines.** We compare against representative state-of-the-art fine-tuning methods: (i) **LoRA** (Hu et al., 2022), the leading PEFT approach that reduces optimizer and gradient memory but still suffers from activation overhead; (ii) **MeZO** (Malladi et al., 2024b), a pioneering ZO method that is memory-efficient but converges slowly; (iii) advanced ZO optimizers such as **HiZOO** (Zhao et al., 2025), which leverages second-order (Hessian) information, and **FZOO** (Dang et al., 2025), designed for faster convergence.

### 4.2 EXPERIMENTS ON ENCODER-BASED MODELS

Table 1: Experiments on RoBERTa-large (350M parameters, k=512). k means that only k samples are taken from each category as the training set.

| Task Type | SST-2 | SST-5 | SNLI | MNLI | RTE | TREC | Average |
|---|---|---|---|---|---|---|---|
| Zero-shot | 79.0 | 35.5 | 50.2 | 48.8 | 51.4 | 32.0 | 49.5 |
| LP | 91.3 | 51.7 | 80.9 | 71.5 | 73.1 | 89.4 | 76.3 |
| FT | 91.9 | 47.5 | 77.5 | 70.0 | 66.4 | 85.0 | 73.1 |
| FT (LoRA) | 91.4 | 46.7 | 74.9 | 67.7 | 66.1 | 82.7 | 71.6 |
| MeZo | 93.3 | **53.2** | 83.0 | 78.3 | 78.6 | 94.3 | 80.1 |
| MeZo (LoRA) | 90.5 | 45.4 | 64.6 | 62.1 | 61.1 | 80.8 | 67.4 |
| HiZOO (LoRA) | 91.7 | 45.3 | 76.5 | 63.1 | 70.4 | 85.6 | 72.1 |
| Ours (k=512) | 94.15 | 36.6 | 85.34 | 78.48 | 75.81 | 97.8 | 78.03 |
| Ours (Full Dataset) | **94.84** | 42.6 | **90.45** | **85.64** | **81.95** | **98** | **82.25** |

We first evaluate DASP on **RoBERTa-large**, with results summarized in Table 1. DASP consistently outperforms all baselines under both many-shot (Full Dataset) and few-shot ($k = 512$) settings. On full-data fine-tuning, it achieves a new state-of-the-art average score of **82.25%**, surpassing Full Fine-tuning (FT) by **+9.15%** and decisively outperforming the strongest ZO baseline, MeZO. These results confirm that DASP not only closes but also exceeds the FO–ZO performance gap while requiring only a fraction of the resources.

In the few-shot ($k = 512$) scenario, DASP maintains strong sample efficiency, reaching **77.98%** average accuracy. Remarkably, this exceeds the performance of FT and FT (LoRA) trained with the full dataset, and remains highly competitive with the full-data MeZO baseline.

## 4.3 EXPERIMENTS ON DECONDER-ONLY LLMS

Table 2: Experiments on three different models (with 1000 examples): Classification (SST-2, RTE, CB, BoolQ, WSC, WIC, MultiRC); Multiple Choice (COPA). Ours method use full datasets.

| Model | Method | SST-2 | RTE | CB | BoolQ | WSC | WIC | MultiRC | COPA | Average |
|-------|--------|-------|-----|-----|-------|-----|-----|---------|------|---------|
| Phi-2 | MeZO | 86.6 | 67.1 | 75.0 | 72.4 | 59.6 | 54.4 | 78.2 | 86 | 72.4125 |
| | HiZOO-L | 88.9 | 68.9 | 75.2 | 72.0 | 62.4 | 59.2 | 79.2 | 86 | 73.975 |
| | FZOO | 87.4 | **70.4** | 83.9 | 79.3 | 61.5 | 56.7 | **81.3** | 86 | 75.8125 |
| | Ours | **94.5** | 64.62 | **85.71** | **84.19** | **63.46** | **63.48** | 80.4 | **86** | **77.795** |
| Llama3-8B | MeZO | 92.2 | 74.4 | 69.6 | 76.7 | 63.5 | 57.8 | 77.6 | 88 | 74.975 |
| | HiZOO-L | 94.3 | 75.1 | 69.6 | 77.1 | 63.5 | 57.7 | 77.9 | 89 | 75.525 |
| | FZOO | 94.3 | 77.6 | 69.6 | 81.8 | 65.4 | 60.8 | 81.5 | 88 | 77.375 |
| | Ours | **95.76** | **84.12** | **83.93** | **89.42** | **78.85** | **71.16** | **85.89** | **95** | **85.51625** |
| OPT-13B | MeZO | 91.4 | 66.1 | 66.0 | 67.6 | 63.5 | 59.4 | 57.3 | 88 | 69.9125 |
| | HiZOO-L | 92.1 | 66.0 | 67.9 | 66.6 | 65.4 | 59.4 | 61.1 | 89 | 70.9375 |
| | FZOO | 93.7 | 71.1 | 69.6 | 72.2 | 63.5 | 60.5 | 66.0 | 87 | 72.95 |
| | Ours | **96.22** | **80.87** | **73.21** | **84.68** | **66.38** | **67.4** | **73.74** | **91** | **79.1875** |

On larger decoder-only models, results in Table 2 demonstrate that DASP consistently establishes a new state-of-the-art across all tasks. For **Llama3-8B**, DASP achieves an average score of **85.52%**, an absolute improvement of over **8.1 points** compared to the next best method, FZOO. Similarly, on **OPT-13B**, it reaches **79.19%**, outperforming the strongest baseline by more than **6.2 points**. This dominant performance highlights the advantage of pre-computed sensitive subspaces over the stochastic search strategies of ZO methods.

Even on the smaller **Phi-2** model, where competition is tighter, DASP still secures the highest overall average score. Although FZOO performs well on specific tasks (e.g., RTE, MultiRC), DASP consistently excels across the full task suite, with particularly notable gains on SST-2 and BoolQ.

## 4.4 ANALYSIS OF MEMORY AND COMPUTATIONAL COSTS

**Data Efficiency and Sensitivity to** $k$**.** We vary the number of samples per class ($k = 16$ to $512$) to test data efficiency. As shown in Table 3, performance saturates quickly: with only **64 samples**, DASP reaches 80.4% accuracy, statistically indistinguishable from full-data fine-tuning.

Table 3: Data efficiency of DASP. Final validation accuracy (%) across datasets when fine-tuned with varying numbers of samples per class ($k$).

| Samples per Class ($k$) | SST-2 | CB | TREC | RTE | COPA |
|-------------------------|-------|-----|------|-----|------|
| 16 | 90.14 | 69.64 | 80.40 | 51.99 | 89.00 |
| 32 | 86.93 | 78.57 | 87.60 | 59.57 | 90.00 |
| 64 | 89.33 | 80.36 | 89.40 | 59.93 | 91.00 |
| 128 | 91.28 | 78.57 | 94.80 | 63.90 | 92.00 |
| 256 | 92.89 | 78.57 | 97.40 | 68.95 | 95.00 |
| 512 | 93.92 | 78.57 | 97.20 | 79.42 | 95.00 |
| Full Dataset | 94.50 | 80.36 | 97.40 | 83.39 | 95.00 |

**Computational Speed.** As shown in Table 4, DASP is also much faster per step than ZO baselines. On OPT-13B, our base implementation runs up to 3× faster than MeZO (0.3766s vs. 1.1108s) since gradients are computed only for a small $r \times r$ core matrix, avoiding costly full-parameter perturbations. Even with the FBP mechanism enabled, DASP remains substantially faster (0.4377s vs. 1.1108s on OPT-13B), showing that its memory savings are achieved with minimal time overhead.

Table 4: Wall-clock time per step for MeZO, HiZOO, HiZOO-L, and Ours, measured on SST-2, averaged over 100 steps. "BS" denotes batch size, "F" indicates FBP, and "C" denotes checkpointing.

| Method | RoBERTa-large(350M) | Phi-2(2.7B) | Llama3(8B) | OPT(13B) |
|---|---|---|---|---|
| MeZO | 0.2092s(BS=64) | 0.3011s(BS=16) | 0.7471s(BS=16) | 1.1108s(BS=16) |
| HiZOO | 0.3023s(BS=64) | 0.4486s(BS=16) | 1.1090s(BS=16) | 1.5225s(BS=16) |
| HiZOO-L | 0.3193s(BS=64) | 0.4851s(BS=16) | 1.1996s(BS=16) | 1.6422s(BS=16) |
| Ours(Without F&C) | 0.1926s(BS=64) | 0.2384s(BS=16) | 0.3315s(BS=16) | 0.3766s(BS=16) |
| Ours(With F) | 0.3854s(BS=64) | 0.2912s(BS=16) | 0.4100s(BS=16) | 0.4377s(BS=16) |
| Ours(With C) | 0.2283s(BS=64) | 0.3021s(BS=16) | 0.4268s(BS=16) | 0.5069s(BS=16) |
| Ours(With F&C) | 0.4292s(BS=64) | 0.3502s(BS=16) | 0.4877s(BS=16) | 0.5721s(BS=16) |

**Memory Footprint.** Table 5 shows peak memory usage across model scales. On the 13B model, DASP requires only 29GB—comparable to MeZO (26GB) and HiZOO-L (29GB), but far lower than Adam(FT) (316GB) and HiZOO (53GB). This efficiency comes from minimizing optimizer state memory to near zero.

Table 5: Peak memory usage on the MultiRC dataset (average sequence length = 400 tokens).

| Model Size | MeZO | HiZOO | HiZOO-L | ICL | Adam(FT) | Ours | Ours(With checkpoint) |
|---|---|---|---|---|---|---|---|
| 1.3B | 4GB | 7GB | 4GB | 6GB | 27GB | 4GB | 3GB |
| 2.7B | 7GB | 13GB | 8GB | 8GB | 55GB | 8GB | 6GB |
| 6.7B | 14GB | 29GB | 15GB | 16GB | 156GB | 16GB | 13GB |
| 13B | 26GB | 53GB | 29GB | 29GB | 316GB | 29GB | 25GB |

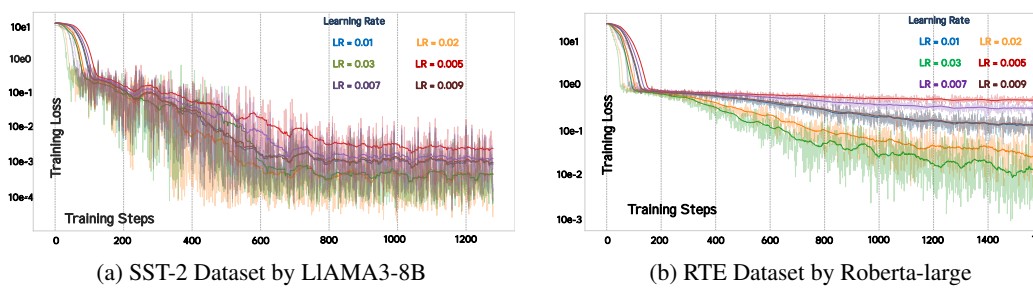

(a) SST-2 Dataset by LlAMA3-8B      (b) RTE Dataset by Roberta-large

Figure 4: Learning rate sensitivity of DASP fine-tuning by Different Models.

**Learning Rate Robustness.** Figure 4 shows DASP's loss curves on SST-2 and RTE under different learning rates. While convergence speed and noise vary, DASP remains stable across 0.005–0.03, without divergence or collapse. This indicates that DASP is robust to learning-rate choices and requires little hyperparameter tuning.

## 5 CONCLUSION

We presented DASP, a novel fine-tuning framework that resolves the long-standing trilemma of performance, memory, and computational cost in adapting Large Language Models. DASP decouples subspace discovery from optimization: an offline ZO-inspired stage efficiently identifies a transferable, task-agnostic low-rank subspace, while the online stage fine-tunes only a small core matrix. Our FBP algorithm further eliminates the activation memory bottleneck for long sequences. As a result, DASP consistently achieves FO-level performance at a resource cost even lower than ZO baselines. Experiments demonstrate that DASP not only bridges but surpasses existing paradigms, offering a practical and scalable solution for the future of LLM fine-tuning.

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

# A APPENDIX: CONVERGENCE PROOF FOR ZEROTH-ORDER STOCHASTIC GRADIENT DESCENT ON NON-CONVEX SMOOTH FUNCTIONS

## A.1 PROBLEM FORMULATION AND ALGORITHM

We consider the unconstrained optimization problem:

$$\min_{x \in \mathbb{R}^p} f(x)$$

where the objective function $f(x)$ is not necessarily convex. We assume we only have zeroth-order access to the function, meaning we can query $f(x)$ for any $x$ but cannot compute its gradient $\nabla f(x)$ directly.

The optimization is performed using a stochastic zeroth-order algorithm based on the two-point gradient estimator. The update rule at each iteration $k$ is given by:

$$x_{k+1} = x_k - \eta G_f^{(2)}(x_k; r, z_k) \tag{16}$$

where $\eta > 0$ is the learning rate (step size), $r > 0$ is a fixed smoothing radius, and $z_k$ is a random vector drawn uniformly from the unit sphere $\mathbb{S}_{p-1} = \{u \in \mathbb{R}^p : ||u|| = 1\}$. The two-point gradient estimator is defined as:

$$G_f^{(2)}(x_k; r, z_k) = \frac{p}{2r} \left( f(x_k + rz_k) - f(x_k - rz_k) \right) z_k \tag{17}$$

For simplicity in the notation that follows, we will denote $G_k = G_f^{(2)}(x_k; r, z_k)$.

## A.2 ASSUMPTIONS

Our proof relies on the following standard assumptions.

**Assumption 1 (L-smoothness).** The function $f$ is differentiable and its gradient $\nabla f$ is Lipschitz continuous with constant $L > 0$. This means for any $x, y \in \mathbb{R}^p$:

$$||\nabla f(x) - \nabla f(y)|| \leq L||x - y||$$

A direct consequence of L-smoothness is the descent lemma:

$$f(y) \leq f(x) + \langle \nabla f(x), y - x \rangle + \frac{L}{2}||y - x||^2$$

**Assumption 2 (Bounded Below).** The function $f$ is bounded below, i.e., there exists a value $f^* = \inf_{x \in \mathbb{R}^p} f(x)$ such that $f(x) \geq f^*$ for all $x$.

## A.3 KEY LEMMAS

We leverage several key properties of the gradient estimator and the smoothed function $f_r(x) = \mathbb{E}_{y \sim \mathbb{B}_p}[f(x + ry)]$, where $\mathbb{B}_p$ is the unit ball in $\mathbb{R}^p$. These are established in the provided reference material.

**Lemma 1 (Expectation of the Gradient Estimator).** The conditional expectation of the gradient estimator $G_k$, given the history $\mathcal{F}_k = \sigma(x_0, ..., x_k)$, is the gradient of the smoothed function $f_r$.

$$\mathbb{E}[G_k|\mathcal{F}_k] = \nabla f_r(x_k)$$

**Lemma 2 (Gradient Difference Bound).** The gradient of the smoothed function is close to the true gradient. The difference is bounded by the smoothing radius $r$.

$$||\nabla f_r(x) - \nabla f(x)|| \leq Lr$$

**Lemma 3 (Rigorous Version).** The conditional second moment of the gradient estimator is bounded. For $z_k \sim \text{Unif}(\mathbb{S}_{p-1})$:

$$\mathbb{E}[||G_k||^2|\mathcal{F}_k] \leq 2p||\nabla f_r(x_k)||^2 + \frac{r^2 L^2 p^2}{2} \tag{18}$$

### A.4 MAIN PROOF OF CONVERGENCE

Our goal is to show that the algorithm converges to a stationary point, which for non-convex optimization means showing that the expected squared norm of the gradient vanishes, i.e., $\frac{1}{K} \sum_{k=0}^{K-1} \mathbb{E}[||\nabla f(x_k)||^2] \to 0$ as $K \to \infty$.

**Step 1: Apply the Descent Lemma.** We start from the descent lemma (a consequence of Assumption 1) applied to $x_{k+1}$ and $x_k$:

$$f(x_{k+1}) \leq f(x_k) + \langle \nabla f(x_k), x_{k+1} - x_k \rangle + \frac{L}{2}||x_{k+1} - x_k||^2 \tag{19}$$

Substitute the update rule $x_{k+1} - x_k = -\eta G_k$:

$$f(x_{k+1}) \leq f(x_k) - \eta \langle \nabla f(x_k), G_k \rangle + \frac{L\eta^2}{2}||G_k||^2 \tag{20}$$

**Step 2: Take Total Expectation.** Now, we take the total expectation over all sources of randomness up to iteration $k+1$. We use the law of total expectation, $\mathbb{E}[X] = \mathbb{E}[\mathbb{E}[X|\mathcal{F}_k]]$.

$$\mathbb{E}[f(x_{k+1})] \leq \mathbb{E}[f(x_k)] - \eta \mathbb{E}[\langle \nabla f(x_k), G_k \rangle] + \frac{L\eta^2}{2}\mathbb{E}[||G_k||^2] \tag{21}$$

Let's analyze the two expectation terms on the right-hand side separately.

**Step 3: Bound the Inner Product Term.** For the inner product term, we first take conditional expectation on $\mathcal{F}_k$. Since $\nabla f(x_k)$ is fixed given $\mathcal{F}_k$, we have:

$$\mathbb{E}[\langle \nabla f(x_k), G_k \rangle | \mathcal{F}_k] = \langle \nabla f(x_k), \mathbb{E}[G_k|\mathcal{F}_k] \rangle$$

Using Lemma 1, this becomes:

$$\langle \nabla f(x_k), \nabla f_r(x_k) \rangle$$

We can express this inner product using the polarization identity:

$$\langle \nabla f(x_k), \nabla f_r(x_k) \rangle = \frac{1}{2} \left( ||\nabla f(x_k)||^2 + ||\nabla f_r(x_k)||^2 - ||\nabla f(x_k) - \nabla f_r(x_k)||^2 \right)$$

Since $||\nabla f_r(x_k)||^2 \geq 0$, we have the lower bound:

$$\langle \nabla f(x_k), \nabla f_r(x_k) \rangle \geq \frac{1}{2} \left( ||\nabla f(x_k)||^2 - ||\nabla f(x_k) - \nabla f_r(x_k)||^2 \right)$$

Now, using Lemma 2, we know $||\nabla f(x_k) - \nabla f_r(x_k)|| \leq Lr$, so:

$$\langle \nabla f(x_k), \nabla f_r(x_k) \rangle \geq \frac{1}{2} \left( ||\nabla f(x_k)||^2 - L^2 r^2 \right)$$

Taking conditional expectation (which is already conditioned on $\mathcal{F}_k$) and then total expectation, we get:

$$\mathbb{E}[\langle \nabla f(x_k), G_k \rangle] = \mathbb{E}[\langle \nabla f(x_k), \nabla f_r(x_k) \rangle] \geq \frac{1}{2}\mathbb{E}[||\nabla f(x_k)||^2] - \frac{1}{2}L^2 r^2$$

Therefore, for the inner product term in the descent inequality, we have:

$$-\eta \mathbb{E}[\langle \nabla f(x_k), G_k \rangle] \leq -\frac{\eta}{2}\mathbb{E}[||\nabla f(x_k)||^2] + \frac{\eta L^2 r^2}{2} \tag{22}$$

**Step 4: Bound the Second Moment Term.** For the second moment term, we take the total expectation of the bound in Lemma 3:

$$\mathbb{E}[||G_k||^2] = \mathbb{E}[\mathbb{E}[||G_k||^2|\mathcal{F}_k]] \leq \mathbb{E}\left[ 2p||\nabla f_r(x_k)||^2 + \frac{r^2 L^2 p^2}{2} \right]$$

Now we need to relate $||\nabla f_r(x_k)||^2$ to $||\nabla f(x_k)||^2$. Using Lemma 2 and the triangle inequality:

$$||\nabla f_r(x_k)|| \leq ||\nabla f(x_k)|| + Lr \Rightarrow ||\nabla f_r(x_k)||^2 \leq 2||\nabla f(x_k)||^2 + 2L^2 r^2$$

Substituting this bound:

$$\mathbb{E}[||G_k||^2] \le 2p\mathbb{E}[2||\nabla f(x_k)||^2 + 2L^2 r^2] + \frac{r^2 L^2 p^2}{2} = 4p\mathbb{E}[||\nabla f(x_k)||^2] + 4pL^2 r^2 + \frac{r^2 L^2 p^2}{2}$$

Therefore,

$$\frac{L\eta^2}{2}\mathbb{E}[||G_k||^2] \le 2L\eta^2 p\mathbb{E}[||\nabla f(x_k)||^2] + 2L^3\eta^2 pr^2 + \frac{L^3\eta^2 r^2 p^2}{4} \tag{23}$$

**Step 5: Combine the Bounds.** Now we substitute the bounds from Eq. (7) and Eq. (8) back into Eq. (6):

$$\mathbb{E}[f(x_{k+1})] \le \mathbb{E}[f(x_k)] - \frac{\eta}{2}\mathbb{E}[||\nabla f(x_k)||^2] + \frac{\eta L^2 r^2}{2} + 2L\eta^2 p\mathbb{E}[||\nabla f(x_k)||^2] + 2L^3\eta^2 pr^2 + \frac{L^3\eta^2 r^2 p^2}{4}$$

Group the terms involving the gradient norm:

$$\mathbb{E}[f(x_{k+1})] \le \mathbb{E}[f(x_k)] - \eta\left(\frac{1}{2} - 2L\eta p\right)\mathbb{E}[||\nabla f(x_k)||^2] + \eta L^2 r^2\left(\frac{1}{2} + 2L\eta p + \frac{L\eta p^2}{4}\right)$$

To ensure convergence, the coefficient of the gradient norm term must be positive. We require $\frac{1}{2} - 2L\eta p > 0$, which implies we must choose a learning rate $\eta < \frac{1}{4Lp}$. Let's set $\eta = \frac{c}{4Lp}$ for some constant $c \in (0,1)$. Then the coefficient becomes $\frac{1}{2} - 2L\left(\frac{c}{4Lp}\right)p = \frac{1-c}{2}$.

Rearrange the inequality to isolate the gradient term on the left side:

$$\eta\left(\frac{1}{2} - 2L\eta p\right)\mathbb{E}[||\nabla f(x_k)||^2] \le \mathbb{E}[f(x_k)] - \mathbb{E}[f(x_{k+1})] + \eta L^2 r^2\left(\frac{1}{2} + 2L\eta p + \frac{L\eta p^2}{4}\right)$$

**Step 6: Sum Over Iterations (Telescoping Sum).** Sum the inequality from $k = 0$ to $K - 1$:

$$\sum_{k=0}^{K-1} \eta\left(\frac{1}{2} - 2L\eta p\right)\mathbb{E}[||\nabla f(x_k)||^2] \le \sum_{k=0}^{K-1}(\mathbb{E}[f(x_k)] - \mathbb{E}[f(x_{k+1})]) + \sum_{k=0}^{K-1} \eta L^2 r^2\left(\frac{1}{2} + 2L\eta p + \frac{L\eta p^2}{4}\right)$$

The first term on the right is a telescoping sum:

$$\sum_{k=0}^{K-1}(\mathbb{E}[f(x_k)] - \mathbb{E}[f(x_{k+1})]) = \mathbb{E}[f(x_0)] - \mathbb{E}[f(x_K)]$$

Since $f$ is bounded below by $f^*$ (Assumption 2), we have $\mathbb{E}[f(x_K)] \ge f^*$. Thus:

$$\mathbb{E}[f(x_0)] - \mathbb{E}[f(x_K)] \le f(x_0) - f^*$$

The second term on the right is a sum of constants:

$$\sum_{k=0}^{K-1} \eta L^2 r^2\left(\frac{1}{2} + 2L\eta p + \frac{L\eta p^2}{4}\right) = K\eta L^2 r^2\left(\frac{1}{2} + 2L\eta p + \frac{L\eta p^2}{4}\right)$$

Combining these, we get:

$$\eta\left(\frac{1}{2} - 2L\eta p\right)\sum_{k=0}^{K-1}\mathbb{E}[||\nabla f(x_k)||^2] \le f(x_0) - f^* + K\eta L^2 r^2\left(\frac{1}{2} + 2L\eta p + \frac{L\eta p^2}{4}\right)$$

**Step 7: Derive the Final Convergence Rate.** Finally, divide by $K$ and the coefficient of the sum to get the average squared gradient norm:

$$\frac{1}{K}\sum_{k=0}^{K-1}\mathbb{E}[||\nabla f(x_k)||^2] \le \frac{f(x_0) - f^*}{K\eta\left(\frac{1}{2} - 2L\eta p\right)} + \frac{L^2 r^2\left(\frac{1}{2} + 2L\eta p + \frac{L\eta p^2}{4}\right)}{\frac{1}{2} - 2L\eta p}$$

Substitute $\eta = \frac{c}{4Lp}$ and simplify:

$$\frac{1}{K}\sum_{k=0}^{K-1}\mathbb{E}[||\nabla f(x_k)||^2] \leq \frac{f(x_0) - f^*}{K \cdot \frac{c}{4Lp} \cdot \frac{1-c}{2}} + \frac{L^2 r^2 \left(\frac{1}{2} + 2L \cdot \frac{c}{4Lp} \cdot p + \frac{L \cdot \frac{c}{4Lp} \cdot p^2}{4}\right)}{\frac{1-c}{2}}$$

$$= \frac{8Lp(f(x_0) - f^*)}{c(1-c)K} + \frac{2L^2 r^2 \left(\frac{1}{2} + \frac{c}{2} + \frac{cp}{16}\right)}{1-c}$$

$$= \frac{8Lp(f(x_0) - f^*)}{c(1-c)K} + \frac{L^2 r^2}{1-c}\left(1 + c + \frac{cp}{8}\right)$$

This expression shows that as the number of iterations $K \to \infty$, the first term goes to zero. The algorithm converges to a region whose size is determined by the second term, which depends on the smoothing radius $r$. To achieve true convergence to a stationary point, one would need to use a decaying radius $r_k \to 0$. For a fixed $r$, the result shows convergence to a neighborhood of a stationary point.

## B  THE USE OF LARGE LANGUAGE MODELS (LLMS)

In the preparation of this paper, Large Language Models (LLMs) were employed solely for language polishing purposes. Specifically, the LLM was used to:

- Improve sentence fluency and readability
- Check for grammatical errors and ensure consistency in expression
- Optimize academic writing style

