# OpenReview forum: "Bridging the Gap Between Zeroth-Order and First-Order Fine-Tuning via Dynamic Adaptive Subspace Pre-tuning"
_ICLR.cc/2026/Conference — ICLR 2026 Conference Withdrawn Submission_

### Official Review · Reviewer_LeHt · 2025-10-29

**Soundness:** 2
**Presentation:** 3
**Contribution:** 4
**Rating:** 4
**Confidence:** 5

**Summary:**

This paper introduces DASP, a fine-tuning framework that pre-computes a low-rank "sensitive subspace" offline and then performs memory-efficient first-order optimization within it. On GLUE/SuperGLUE benchmarks, DASP achieves accuracy comparable to full fine-tuning and LoRA while reducing memory usage below ZO methods like MeZO. The proposed Flow Back-Propagation (FBP) algorithm enables constant-memory training for long sequences, and the subspace demonstrates strong transferability across NLP tasks.

**Strengths:**

This paper introduces a well-executed approach to efficient fine-tuning through its decoupled subspace discovery and optimization framework. The method is technically sound, demonstrating clear practical advantages in reducing memory overhead while maintaining competitive performance. The presentation is coherent and accessible, effectively communicating its core contributions.

**Weaknesses:**

1) The title and provided keywords are misleading. The approach involves obtaining the projection space using a zero-order method and updating the embedding space with a first-order method—making it fundamentally a first-order method rather than a zero-order optimization approach. Therefore, in experimental comparisons, it should be compared against first-order methods, such as Galore [1](which performs bilateral projection on gradients and implements layer-wise updates to reduce memory usage). A comprehensive comparison should include convergence rate, memory consumption, and time cost.

2) To solidly demonstrate that the performance gain comes from the quality of the probed subspace rather than merely the existence of a fixed low-rank structure, I strongly recommend adding an ablation study comparing against a baseline that uses a random, fixed subspace (with P and Q being random matrices). This would effectively isolate and highlight the contribution of the novel probing phase.

3) The claim that the sensitive subspace is 'dataset-agnostic' seems too strong. The probing phase uses task data (e.g., SST-2) to compute the pseudo-gradients. Would the subspace be different if probed on a completely different type of data (e.g., code or audio)? The consistency shown across time and layers only proves robustness to different batches from a similar data distribution, not agnosticity. More empirical evidence is required to solidly support this claim.

[1] Zhao, Jiawei, et al. "GaLore: Memory-Efficient LLM Training by Gradient Low-Rank Projection." Forty-first International Conference on Machine Learning.

**Questions:**

1) Would it be possible to include a comparison with a baseline that uses a fixed random subspace (i.e., P and Q as random matrices)? This would help better isolate the specific contribution of the probing phase from the general benefits of using a low-rank structure.

2) The method currently employs a single fixed subspace identified during probing, based on the assumption that a stable "sensitive subspace" exists. It would be interesting to investigate whether the sensitive subspace might shift during optimization. Could the authors explore a more dynamic strategy—for instance, adapting the subspace at different fine-tuning stages? An ablation comparing fixed versus periodically updated subspaces would offer valuable insights into the framework's adaptability.

3) As the probing phase itself utilizes task data, could the authors provide additional evidence—such as cross-domain transfer results (e.g., probing on text and fine-tuning on code)—to further support the intriguing claim that the subspace is model-intrinsic and data-agnostic?

4) A direct comparison with recent gradient-based low-rank ZO methods such as SubZero[1] and LoZO[2] would be highly valuable, especially regarding convergence speed and total training time overhead, to better understand the practical trade-offs of DASP.

5) Based on gradient low-rank projection, Galore[3] can even be used for pre-training. It makes one wonder whether DASP could also be applied to large language model pre-training. Perhaps DASP would require certain modifications, such as subspace switching strategies or unilateral projection of gradients.

I will re-score based on the author's rebuttal.

[1] Yu, Ziming, et al. "Zeroth-order fine-tuning of llms in random subspaces." Proceedings of the IEEE/CVF International Conference on Computer Vision. 2025.

[2] Chen, Yiming, et al. "Enhancing Zeroth-order Fine-tuning for Language Models with Low-rank Structures." The Thirteenth International Conference on Learning Representations.

[3] Zhao, Jiawei, et al. "GaLore: Memory-Efficient LLM Training by Gradient Low-Rank Projection." Forty-first International Conference on Machine Learning.

**Details Of Ethics Concerns:**

None.

---

### Official Review · Reviewer_rwWc · 2025-10-29

**Soundness:** 2
**Presentation:** 2
**Contribution:** 2
**Rating:** 2
**Confidence:** 5

**Summary:**

The paper studies the problem of efficient fine-tuning of LLMs. While first-order methods are the standard, they consume a significant amount of memory for large models with large sequence length or batch-size. Although zeroth-order methods can greatly reduce the memory usage by avoiding backpropagation, their performance are usually worse than first-order methods. This paper proposes a way to use similar methods as zeroth-order optimization in the first stage to find the subspace that leads to largest performance change, and then fixes this subspace and only tune a much smaller matrix in the second stage with first-order methods. Experiments show that this new method has similar memory footprint as zeroth-order optimization, while achieving similar or even better performance as first-order optimization.

**Strengths:**

The paper is in general clearly written and self-contained. The studied problem is well-explained and also well-motivated. The topic of memory efficient fine-tuning of LLMs is interesting and relevant to the community. The proposed algorithm is explained in detail and makes sense.

**Weaknesses:**

1. I think there is a huge body of related works that are not properly discussed in the paper. I suggest the authors to provide a better literature review to place their work. The parameterization of $W'=W+P\delta Q^\top$ is not new; see for example [1,2,3,4,5]. The main difference between this work and these previous works is how $P$ and $Q$ are updated. In [2,3], $P$ and $Q$ are also updated using Riemannian optimization. The most relevant papers to this work are [4,5], where $P$ and $Q$ are fixed as SVD of the pretrained weights, and only $\delta$ is updated as well. This choice of subspace is well-motivated and also extensively used in other works; see [6,7]. LoRA initialization has also been studied to not start from purely random; see [6,8]. Other types of subspace optimization based on the intuition of low-rank gradient also exist in [9,10].

2. Stage 1 is not well-discussed in experiments. In particular, how is the calibration datasets $D_{cal}$ selected, and what is the size of it? Will the performance largely depend on how good this dataset is? Moreover, I don't think the memory and running time of stage 1 are discussed? How large is the total iteration number $K$ for updating $P$ and $Q$? Considering that this is the major difference compared to previous work, this part desires more discussions and justification, and more ablation study on how $P$ and $Q$ affect later training is required.

3. I find that the reported baselines in Table 1 and 2 are significantly lower than that in orther paper. For example, MeZO reports that the performance on RoBERTa-large with $k=512$ is 93.9, 55.9, 88.7, 84.4, 82.7, 97.4 for FT and 94.2, 55.3, 88.3, 83.9, 83.2, 97.0 for FT (LoRA). Both are much larger than the numbers presented in Table 1 and are potentially better than the proposed method. Actually, I find that the reported numbers in Table 1 is the performance of $k=16$ in MeZO paper. **This is a very severe issue about research integrity, which is super misleading and gives wrong conclusions**. The same happens to Table 2 as well. In addition, why are FT and FT (LoRA) missing in Table 2?

[1] AdaLoRA: Adaptive Budget Allocation for Parameter-Efficient Fine-Tuning. ICLR 2023 (arXiv: 2303.10512)

[2] StelLA: Subspace Learning in Low-rank Adaptation using Stiefel Manifold. NeurIPS 2025 (arXiv: 2510.01938)

[3] PoLAR: Polar-Decomposed Low-Rank Adapter Representation. 2025. (arXiv: 2506.03133)

[4] LoRA-XS: Low-Rank Adaptation with Extremely Small Number of Parameters. 2024. (arXiv:2405.17604)

[5] Initialization using Update Approximation is a Silver Bullet for Extremely Efficient Low-Rank Fine-Tuning. 2024. (arXiv: 2411.19557)

[6] PiSSA: Principal Singular Values and Singular Vectors Adaptation of Large Language Models. NeurIPS 2024. (arXiv: 2404.02948)

[7] SVFT: Parameter-Efficient Fine-Tuning with Singular Vectors. NeurIPS 2024 (arXiv:2405.19597)

[8] On the Crucial Role of Initialization for Matrix Factorization. ICLR 2025 (arXiv: 2410.18965)

[9] GaLore: Memory-Efficient LLM Training by Gradient Low-Rank Projection. ICML 2024 (arXiv: 2403.03507)

[10] Memory-Efficient LLM Training with Online Subspace Descent. NeurIPS 2024 (arXiv: 2408.12857)

**Questions:**

Major questions are asked in weaknesses. Other questions.

1. In Figure 2(c), it is strange that MeZO on LoRA has very good performance at the beginning, but the accuracy drops after training. This perhaps implies that MeZO baseline is not well-tuned. In particular, I find that this figure uses larger stepsizes 0.001 for MeZO (LoRA) compared to 0.0001 for LoRA. This does not follow the usual standard that selects smaller stepsizes for MeZO; see the MeZO paper.

2. I am a bit unsure why stage 1 converges. Can authors provide more insights and explanations on where it converges to? Will it converge to the solutions of (3) and why?

3. Why does the MeZO references appear twice in line 520-525?

---

### Official Review · Reviewer_BLTH · 2025-10-30

**Soundness:** 2
**Presentation:** 2
**Contribution:** 2
**Rating:** 2
**Confidence:** 3

**Summary:**

The paper proposes DASP, a two‑stage framework intended to combine the memory advantages of ZO methods with the accuracy of FO fine‑tuning. Stage 1 aims to discover, per linear layer, fixed orthonormal bases $P\in\mathbb{R}^{m\times r}$ and $Q\in\mathbb{R}^{n\times r}$ that define a “sensitive” low‑rank subspace by maximizing loss deviation under random rank‑(r) perturbations on a calibration set. Stage 2 then fine‑tunes only a tiny core matrix $\delta\in\mathbb{R}^{r\times r}$ in the update $W' = W + P\delta Q^\top$, with a projection‑aware Flow Backpropagation scheme that streams the backward pass over sequence chunks to make activation memory near independent of sequence length while keeping the gradient for $\delta$ exact, $\nabla_\delta \mathcal{L}=(XQ)^\top(GP)$. Experiments across RoBERTa‑large, Phi‑2, Llama‑3‑8B, and OPT‑13B are done.

**Strengths:**

1. The derivation $\nabla_\delta \mathcal{L}=(XQ)^\top(GP)$ is sound, and the block‑sum identity (Eqs. 13–15) justifies exact chunked accumulation, which helps streaming FBP with memory that depends on chunk size C and rank r, not full sequence length T.
2. Simplicity. Updating only an $r\times r$ core $\delta$ while freezing $P,Q$ is elegant and potentially yields tiny optimizer state compared to LoRA’s $A,B$.
3. Figure 3 shows consistent low‑rank spectral decay of per‑layer gradient projections, supporting the premise that a compact subspace may suffice for adaptation (though broader validation would help).

**Weaknesses:**

1. Algorithm 1 chooses a probe $\omega^*$ and then updates $P \leftarrow \mathrm{orth}(PU)$, $Q \leftarrow \mathrm{orth}(QV)$, which cannot change the column spaces; it only rotates bases inside the same $r$-dimensional span (Eq. 4–5 in Algorithm 1). Consequently, the method cannot move toward a “more sensitive” subspace than the random initialization, contradicting the stated goal of Eq. 3. This is a major methodological issue.

2. Table 2 title says “with 1000 examples,” yet the caption ends with “Ours method use full datasets.” That implies a data‑budget mismatch favoring DASP.
Table 1 mixes “Ours (k=512)” and “Ours (Full Dataset)” without showing matching k‑restricted rows for FO baselines (FT/LoRA).
Fig. 2b–c plot loss/accuracy vs “cumulative forward passes.” That metric disadvantages FO methods (where an update cost includes backward) and obscures time‑to‑accuracy and step‑to‑accuracy comparisons; wall‑clock and step counts should be reported alongside.

3. DASP emphasizes that $P,Q$ are task‑agnostic and reusable, but the paper does not document the calibration set size/content, offline compute, or explicit transfer experiments reusing the same $P,Q$ across tasks.

4. Memory measurement protocol (hardware, precision, CUDA allocator, checkpointing/FBP settings, inclusion of $P,Q$ in “weights,” and KV‑cache handling) is under‑specified in Fig. 2a and Table 5.
No ablation of rank r vs. accuracy/memory, and r is not reported per model/layer in the main tables.
Variance/seed reporting is missing.
Appendix A provides a standard ZO‑SGD convergence sketch unrelated to Algorithm 1 and contains numbering mistakes (Step 5 references the wrong equations).
Table 5 includes an ICL baseline not defined in Sec. 4.1 baselines.

**Questions:**

1.  How do $P,Q$ ever leave the initial span if updates are $P \leftarrow \mathrm{orth}(PU)$, $Q \leftarrow \mathrm{orth}(QV)$ with $U,V\in\mathbb{R}^{r\times r}$? Please either provide an update that can incorporate directions from the ambient space (e.g., accumulate responses in ambient coordinates and re‑estimate the subspace via truncated SVD/power iteration), or prove that within‑span rotations suffice to optimize Eq. 3.

2.  What calibration data $\mathcal{D}_{\text{cal}}$ were used (size, domains), and is one set of $P,Q$ reused across all tasks in Tables 1–2? Please report offline wall‑clock/forward‑pass cost and show a “compute once, reuse many tasks” experiment.

3. In Table 2, were all methods trained on exactly 1000 labeled examples, or did DASP use the full dataset? Please recompute with matched data budgets (and matched training steps) and add step‑to‑accuracy and time‑to‑accuracy plots alongside the “forward‑pass” axis in Fig. 2.

4. What ranks r are used per model/layer? Please add accuracy/memory vs r ablations and clarify whether $P,Q$ storage is included in memory accounting; also state whether KV‑caches are counted.

5. For FBP, what exactly is stored vs recomputed? How does peak memory scale with depth and chunk size C for long contexts?

---

### Official Review · Reviewer_KXa8 · 2025-11-01

**Soundness:** 2
**Presentation:** 3
**Contribution:** 3
**Rating:** 4
**Confidence:** 3

**Summary:**

The paper proposes DASP (Dynamic Adaptive Subspace Pre‑tuning) to reconcile the accuracy of FO fine‑tuning with the memory efficiency of ZO methods for LLM adaptation. DASP decouples subspace discovery from optimization:  (1) an offline, ZO‑inspired probing stage iteratively identifies layer‑wise, low‑rank “sensitive” subspaces by updating orthonormal bases. (2) online stage then restricts updates to a tiny r×r core matrix. To remove the activation bottleneck, the authors introduce Flow Backpropagation (FBP)—an exact streaming backprop algorithm that accumulates projected, chunk‑wise gradients. Experiments report faster convergence and accuracy on par with FO while keeping memory comparable to ZO.

**Strengths:**

1.  It proposes two‑stage paradigm: first discover the adaptation subspace, then optimize within it, so the optimizer/gradient state scales only as r×r by fixing P,Q and training a tiny core matrix δ.
2. It  derives ∇δL=(XQ)⊤(GP) and proves its block‑wise summation equivalence.

**Weaknesses:**

1. The method section notes FBP is “inspired by StreamBP” and hinges on block‑wise summation equivalence (Eq. 15), which is standard linear algebra and also the basis of recent exact streaming backprop works (StreamBP, MsT). The paper does not deliver a comparison against those FO memory‑efficient backprop methods under matched chunk sizes/precisions to prove novelty or superiority.

2. The offline stage requires, for every linear layer, multiple forward probes, SVDs of the best probe, and re‑orthogonalization over K iterations, but the paper gives no GPU‑hours, wall‑times, or K/N_probes/ε settings, nor the size and source of Dcal. Calling this stage “lightweight” is unsubstantiated; for 8B/13B models this could be very costly.

3. This paper claims that the computed subspace bases (P, Q) can be reused across different downstream tasks, amortizing the one‑time cost, but lack evidence.

4. The paper claims “exact long‑context training with near‑constant memory” via FBP, yet the only memory table uses with ~400 tokens, which is not a long‑context regime by LLM standards, and no experiments at 4k/8k/32k tokens are shown.

5. The only hyperparameter exploration shown is a LR sweep (Fig. 4). There is no ablation for: (i) removing FBP; (ii) varying chunk size ; (iii) different calibration set sizes/sources; (iv) reusing vs. recomputing (P,Q); (v) impact of noisy/off‑domain Dcal.
	​

.

**Questions:**

In table 1, are baselines are trained with k=512 or full dataset? If 512, then DASP does not exceed MeZo.

Is there any strategy choice of global rank r and chunk size? without this, practical guidance is missing.

some typos:
“EXPERIMENTS ON DECONDER‑ONLY LLMS” → “DECODER‑ONLY LLMs.”

---

### Note · Authors · 2026-01-26

I have read and agree with the venue's withdrawal policy on behalf of myself and my co-authors.

---

### Meta-Review · Area_Chair_rXvE · 2026-01-07

**Summary:**

This paper proposes DASP (Dynamic Adaptive Subspace Pre-tuning), a two-stage framework that aims to bridge zeroth-order (ZO) and first-order (FO) fine-tuning by first identifying task-sensitive low-rank subspaces via a ZO-inspired probing phase and then performing memory-efficient FO optimization within those fixed subspaces. The method also introduces Flow Backpropagation (FBP) to reduce activation memory. While the concept is promising and the claimed memory-accuracy trade-off is appealing, multiple reviewers raise serious concerns regarding experimental validity, methodological soundness, and missing comparisons, which significantly undermine confidence in the results.

**Reviewer Concerns:**

No rebuttal was submitted, so none of the concerns were addressed. Major outstanding issues include:
(1) Inconsistent or misleading baselines: Reviewer rwWc points out that the reported FT/LoRA numbers in Tables 1–2 appear to match MeZO’s k=512 results—not full-data FO baselines—creating an unfair advantage for DASP.
(2) Methodological flaw in subspace discovery: Reviewer BLTH argues that Algorithm 1 only rotates within the initial random subspace and cannot expand into more sensitive directions, contradicting the optimization goal in Eq. 3.
(3) Missing ablations and details: No analysis of calibration set size/source, offline compute cost, rank sensitivity, or comparison to fixed random subspaces (as suggested by Reviewer LeHt).
(4) Inadequate long-context validation: Claims of “constant-memory long-context training” are unsupported, as experiments use only ~400-token sequences.
(5) Insufficient comparison to recent FO-efficient methods: GaLore, SubZero, LoRA-XS, PiSSA, and other low-rank or memory-efficient FO/ZO hybrids are either omitted or not fairly benchmarked.

**Reviewer Scores:**

Reviewer KXa8: Rating 4 → likely unchanged (weak accept-leaning reject)
Reviewer BLTH: Rating 2 → would likely remain a firm reject due to methodological concerns
Reviewer rwWc: Rating 2 → strong reject, citing potential data integrity issues and missing baselines
Reviewer LeHt: Rating 4

No rebuttal

---

### Decision · Program_Chairs · 2026-01-26

Reject